# HOFAR: High-Order Augmentation of Flow Autoregressive Transformers

## Abstract

Flow Matching and Transformer architectures have demonstrated remarkable performance in image generation tasks, with recent work FlowAR [Ren et al., 2024] synergistically integrating both paradigms to advance synthesis fidelity. However, current FlowAR implementations remain constrained by first-order trajectory modeling during the generation process. This paper introduces a novel framework that systematically enhances flow autoregressive transformers through high-order supervision. We provide theoretical analysis and empirical evaluation showing that our High-Order FlowAR (HOFAR) demonstrates measurable improvements in generation quality compared to baseline models. The proposed approach advances the understanding of flow-based autoregressive modeling by introducing a systematic framework for analyzing trajectory dynamics through high-order expansion.

## 1 Introduction

Recently, flow-matching Lipman et al. (2022) and diffusion models Ho et al. (2020) have demonstrated remarkable capabilities in the field of image generation Rombach et al. (2022); Esser et al. (2024). Several works have explored extending these models to generate images with an additional dimension, such as incorporating a temporal dimension for video generation Singer et al. (2022); Li et al. (2023) or a 3D spatial dimension for 3D object generation Xue et al. (2024); Mo (2024). Even 4D generation Zhang et al. (2025); Liang et al. (2024a) has become feasible using diffusion models. Another prominent line of research focuses on auto-regressive models, where the Transformer framework has achieved groundbreaking success in natural language processing. Models such as GPT-4 Achiam et al. (2023), Gemini 2 Deepmind (2024), and DeepSeek Guo et al. (2025) have significantly impacted millions of users worldwide.

Given the success of the auto-regressive generation paradigm and the Transformer framework, recent works have explored integrating auto-regressive generation into image generation. A representative example is the Visual Auto-Regressive (VAR) model Tian et al. (2025), which introduces hierarchical image generation with different image patches. Other works, such as FlowAR Ren et al. (2024) and ARFlow Hui et al. (2025), integrate flow-matching with auto-regressive generation. However, these existing approaches primarily focus on modeling the direct transition path between the prior distribution and the target image distribution, paying less attention to high-order dynamics. High-order dynamics play a crucial role in capturing complex dependencies between different modalities, which is especially important for tasks like video generation that require long-term coherence. Moreover, high-order supervision enhances a model's generalization ability by encouraging it to learn fundamental generative principles rather than relying on lower-order patterns.

Motivated by these insights, we propose High-Order FlowAR (**HOFAR**), an approach that builds upon the strengths of auto-regressive models and flow-matching techniques while extending them to model higher-order interactions. By explicitly incorporating high-order dynamics, HOFAR improves realism, coherence, and generalization in generative tasks. We theoretically prove that HOFAR maintains computational efficiency compared to its base models while empirically demonstrating its superior performance.

In summary, our contributions are as follows:

- We introduce HOFAR, a novel framework that integrates high-order dynamics into flow-matching-based auto-regressive generation, enhancing the model's ability to capture complex dependencies.

- We provide a theoretical analysis showing that HOFAR maintains computational efficiency while benefiting from high-order modeling.

- We conduct empirical evaluations demonstrating that HOFAR achieves improved generation quality, coherence, and generalization compared to existing auto-regressive generative models.

## 2 PRELIMINARY

In this section, we introduce the formal mathematical definitions for the FlowAR model and our High-Order FlowAR (HOFAR) model. These definitions provide the foundational framework for understanding the preprocessing, downsampling, upsampling, and transformer-based components of the proposed architecture. In Section 2.1, we introduce the notations we used in this work. In Section 2.2, we describe the preprocessing steps applied to input images before they are fed into the model. In Section 2.3, we detail the autoregressive Transformer architecture, which generates the conditional embeddings utilized by the flow-matching components in the FlowAR model.

### 2.1 NOTATIONS

Given a matrix $X \in \mathbb{R}^{hw \times d}$, we denote its tensorized form as $\mathsf{X} \in \mathbb{R}^{h \times w \times d}$. Additionally, we define the set $[n]$ to represent $\{1, 2, \cdots, n\}$ for any positive integer $n$. We define the set of natural numbers as $\mathbb{N} := \{0, 1, 2, \dots\}$. Let $X \in \mathbb{R}^{m \times n}$ be a matrix, where $X_{i,j}$ refers to the element at the $i$-th row and $j$-th column. When $x_i$ belongs to $\{0, 1\}^*$, it signifies a binary number with arbitrary length. In a general setting, $x_i$ represents a length $p$ binary string, with each bit taking a value of either 1 or 0. Given a matrix $X \in \mathbb{R}^{n \times d}$, we define $\|X\|_\infty$ as the maximum norm of $X$. Specifically, $\|X\|_\infty = \max_{i,j} |X_{i,j}|$.

### 2.2 FLOWAR PREPROCESSING PROCESS

We begin by introducing the preprocessing procedure of the FlowAR model. The image is first passed through a Variational Autoencoder (VAE) to obtain a latent image embedding before being processed by the main body of the FlowAR model.

Let $\mathsf{X} \in \mathbb{R}^{h \times w \times c}$ denote the image embedding generated by the VAE, where $h$, $w$, and $c$ represent the height, width, and number of channels, respectively. The next step involves downsampling the image embedding $\mathsf{X}$ to multiple scales. To formalize this process, we first define the linear downsampling function.

**Definition 2.1** (Linear Downsampling Function). *If the following conditions hold:*

- *Let $\mathsf{X} \in \mathbb{R}^{h \times w \times c}$ denote the input tensor, where $h, w, c$ represent height, width, and the number of channels, respectively.*

- *Let the positive integer $r \geq 1$ denote the scaling factor.*

*The linear downsampling function $\phi_{\mathrm{down}}(\mathsf{X}, r)$ computes an output tensor $\mathsf{Y} \in \mathbb{R}^{(h/r) \times (w/r) \times c}$.*

*To be more specific, let $\Phi_{\mathrm{down}} \in \mathbb{R}^{(h/r \cdot w/r) \times hw}$ denote a linear transformation matrix. The downsampling transformation consists of three steps:*

- *Reshape $\mathsf{X}$ into the matrix $X \in \mathbb{R}^{hw \times c}$ by flattening its spatial dimensions.*

- *Apply the linear transformation matrix $\Phi_{\mathrm{down}}$ on $\mathsf{X}$ as*

$$Y = \Phi_{\mathrm{down}} X \in \mathbb{R}^{(h/r \cdot w/r) \times c},$$

- *Reshaped back to $\mathsf{Y} \in \mathbb{R}^{(h/r) \times (w/r) \times c}$.*

Next, we define the multi-scale downsampling tokenizer, which leverages the linear downsampling function to generate a sequence of token maps at multiple scales.

**Definition 2.2** (Multi-Scale Downsampling Tokenizer). *If the following conditions hold:*

- *Let $\mathsf{X} \in \mathbb{R}^{h \times w \times c}$ denote the image embedding generated by VAE.*

- *Let $K \in \mathbb{N}$ denote the number of scales.*

- *Let the positive integer $a \geq 1$ denote the base scaling factor.*

- *For $i \in [K]$, we define scale-specific factors $r_i := a^{K-i}$ and use the linear downsampling function $\phi_{\mathrm{down}}(\mathsf{X}, r_i)$ from Definition 2.1.*

*We define the multi-scale downsampling tokenizer as $\mathsf{TN}(\mathsf{X}) := \{\mathsf{Y}^1, \dots, \mathsf{Y}^K\}$, which outputs a sequence of token maps $\{\mathsf{Y}^2, \mathsf{Y}^2, \dots, \mathsf{Y}^K\}$, where the $i$-th token map is generated by*

$$\mathsf{Y}^i := \phi_{\mathrm{down},i}(\mathsf{X}, r_i) \in \mathbb{R}^{(h/r_i) \times (w/r_i) \times c},$$

During inference, we need to upsample the embeddings after each processing step. To formalize this operation, we define the bicubic upsampling function as follows.

**Definition 2.3** (Upsampling Function). *If the following conditions hold:*

- *Let $\mathsf{X} \in \mathbb{R}^{h \times w \times c}$ denote the input tensor, where $h, w, c$ represent height, width, and the number of channels, respectively.*

- *Let the A positive integer $r \geq 1$ denote the scaling factor.*

- *Let $W : \mathbb{R} \to [0, 1]$ denote the bicubic kernel.*

*We define the bicubic upsampling function as $\phi_{\mathrm{up}}(\mathsf{X}, r)$, which computes $\mathsf{Y} \in \mathbb{R}^{rh \times rw \times c}$. For every output position $i \in [rh], j \in [rw], l \in [c]$:*

$$\mathsf{Y}_{i,j,l} = \sum_{s=-1}^{2} \sum_{t=-1}^{2} W(s) \cdot W(t) \cdot \mathsf{X}_{\lfloor \frac{i}{r} \rfloor + s, \lfloor \frac{j}{r} \rfloor + t, l}$$

## 2.3 AUTOREGRESSIVE TRANSFORMER ARCHITECTURE

The downsampled embeddings are then fed into the transformer architecture to generate the condition tensor for the flow matching model. The autoregressive transformer is a key component of the FlowAR model. Below, we define its attention layer, feedforward layer, and the overall autoregressive transformer.

**Definition 2.4** (Attention Layer). *If the following conditions hold:*

- *Let $\mathsf{X} \in \mathbb{R}^{h \times w \times c}$ denote the input tensor, where $h, w, c$ represent height, width, and the number of channels, respectively.*

- *Let $W_Q, W_K, W_V \in \mathbb{R}^{c \times c}$ denote the weight matrices, which will be used in query, key, and value projection, respectively.*

*The attention layer $\mathsf{Attn}(\mathsf{X})$ is defined by computing the output tensor $\mathsf{Y} \in \mathbb{R}^{h \times w \times c}$ in the following three steps:*

- *Reshape $\mathsf{X}$ into a matrix $X \in \mathbb{R}^{hw \times c}$ with spatial dimensions collapsed.*

- *Attention matrix computation. For $i, j \in [hw]$, compute pairwise scores:*

$$A_{i,j} := \exp(X_{i,*} W_Q W_K^\top X_{j,*}^\top), \ \text{for } i, j \in [hw].$$

- *Normalization. Compute diagnal matrix $D := \mathrm{diag}(A \mathbf{1}_n) \in \mathbb{R}^{hw \times hw}$, where $\mathbf{1}_n$ is the all-ones vector. And compute:*

$$Y := D^{-1} A X W_V \in \mathbb{R}^{hw \times c}.$$

- *Reshape $Y$ to $\mathsf{Y} \in \mathbb{R}^{h \times w \times c}$.*

The feedforward layer is another critical component of the transformer architecture. We define it as follows.

**Definition 2.5** (Feed Forward Layer). *If the following conditions hold:*

- *Let $\mathsf{X} \in \mathbb{R}^{h \times w \times c}$ denote the input tensor, where $h, w, c$ represent height, width, and the number of channels, respectively.*

- *Let $W_1, W_2 \in \mathbb{R}^{c \times d}$ denote the weight matrices and $b_1, b_2 \in \mathbb{R}^{1 \times d}$ denote the bias vectors.*

- *Let $\sigma : \mathbb{R} \to \mathbb{R}$ denote the ReLU activation function which is applied element-wise.*

*We defined the feedforward operation as $\mathsf{Y} := \mathsf{FFN}(\mathsf{X})$.*

*To be more specific, it computes an output tensor $\mathsf{Y} \in \mathbb{R}^{h \times w \times d}$ in the following steps:*

- *Reshape $\mathsf{X}$ into a matrix $X \in \mathbb{R}^{hw \times c}$ with spatial dimensions collapsed.*

- *For each $j \in [hw]$, compute*

$$
Y_{j,*} = \underbrace{X_{j,*}}_{1 \times c} + \sigma( \underbrace{X_{j,*}}_{1 \times c} \cdot \underbrace{W_1}_{c \times c} + \underbrace{b_1}_{1 \times c} ) \cdot \underbrace{W_2}_{c \times c} + \underbrace{b_2}_{1 \times c} \in \mathbb{R}^{1 \times c}
$$

  *where $\sigma$ acts element-wise on intermediate results. Then reshape $Y \in \mathbb{R}^{hw \times c}$ into $\mathsf{Y} \in \mathbb{R}^{h \times w \times c}$.*

Using the attention and feedforward layers, we now define the autoregressive transformer.

**Definition 2.6** (Autoregressive Transformer). *If the following conditions hold:*

- *Let $\mathsf{X} \in \mathbb{R}^{h \times w \times c}$ denote the input tensor, where $h, w, c$ represent height, width, and the number of channels, respectively.*

- *Let $K \in \mathbb{N}$ denote the scale number, which is the number of total scales in FlowAR.*

- *For $i \in [K]$, let $\mathsf{Y}_i \in \mathbb{R}^{(h/r_i) \times (w/r_i) \times c}$ denote the token maps generated by the Multi-Scale downsampling tokenizer defined in Definition 2.2 where $r_i = a^{K-i}$ with base $a \in \mathbb{N}^+$.*

- *For $i \in [K]$, let $\phi_{\mathrm{up},i}(\cdot, a) : \mathbb{R}^{(h/r_i) \times (w/r_i) \times c} \to \mathbb{R}^{(h/r_{i+1}) \times (w/r_{i+1}) \times c}$ denote the upsampling functions as defined in Definition 2.3.*

- *For $i \in [K]$, let $\mathsf{Attn}_i(\cdot) : \mathbb{R}^{(\sum_{j=1}^{i} h/r_j \cdot w/r_j) \times c} \to \mathbb{R}^{(\sum_{j=1}^{i} h/r_j \cdot w/r_j) \times c}$ denote the attention layer which acts on flattened sequences of dimension defined in Definition 2.4.*

- *For $i \in [K]$, let $\mathsf{FFN}_i(\cdot) : \mathbb{R}^{(\sum_{j=1}^{i} h/r_j \cdot w/r_j) \times c} \to \mathbb{R}^{(\sum_{j=1}^{i} h/r_j \cdot w/r_j) \times c}$ denote the feed forward layer which acts on flattened sequences of dimension defined in Definition 2.5.*

- *Let $\mathsf{Z}_{\mathrm{init}} \in \mathbb{R}^{(h/r_1) \times (w/r_1) \times c}$ denote the initial condition embedding which encodes class information.*

*Then, the autoregressive processing is:*

- **Initialization:** *Let $\mathsf{Z}_1 := \mathsf{Z}_{\mathrm{init}}$.*

- **Iterative sequence construction:** *For $i \geq 2$.*

$$
\mathsf{Z}_i := \mathsf{Concat}(\mathsf{Z}_{\mathrm{init}}, \phi_{\mathrm{up},1}(\mathsf{Y}^1, a), \dots, \phi_{\mathrm{up},i-1}(\mathsf{Y}^{i-1}, a))
$$

  *where $\mathsf{Concat}$ reshapes tokens into a unified spatial grid.*

- **Transformer block:** *For $i \in [K]$,*

$$
\mathsf{TF}_i(\mathsf{Z}_i) := \mathsf{FFN}_i(\mathsf{Attn}_i(\mathsf{Z}_i)) \in \mathbb{R}^{(\sum_{j=1}^{i} h/r_j \cdot w/r_j) \times c}
$$

- **Output decomposition:** *Extract the last scale's dimension from the reshaped $\mathsf{TF}_i(\mathsf{Z}_i)$ to generate $\widehat{\mathsf{Y}}_i \in \mathbb{R}^{(h/r_i) \times (w/r_i) \times c}$.*

# 3 MAIN RESULTS

In this section, we present our theoretical analysis of the computational efficiency of the HOFAR model. We demonstrate that despite incorporating high-order dynamics supervision, the increase in computational complexity for both training and inference remains marginal compared to the significant performance improvements achieved.

**Theorem 3.1** (Computational Efficiency of HOFAR). *In accordance with Definition 2.6, the auto-regressive Transformer architecture incorporates $m$ attention layers. The image input $x_{\text{img}} \in \mathbb{R}^{n \times n \times c}$ is encoded with $n^2$ spatial units, $c$ channels, and a $d$-dimensional latent representation. The HOFAR model demonstrates computational costs of $O(kmn^4d^2)$ for both training and inference under the specified structural constraints.*

*Proof.* The proof follows from Lemma 4.3 and Lemma 4.4. □

---

**Algorithm 1** High-Order FlowAR Training

---

1: **procedure** HOFARTRAINING($\theta, D$)
2:     /* $\theta$ denotes the model parameters of TF, $\text{FM}_{\text{first}}$, $\text{FM}_{\text{second}}$ */
3:     /* $D$ denotes the training dataset. */
4:     **while** not converged **do**
5:         /* Sample an image from dataset. */
6:         $x_{\text{img}} \sim D$
7:         /* Init loss as 0. */
8:         $\ell \leftarrow 0$
9:         /* Train the model on $K$ pyramid layers. */
10:         **for** $i = 1 \rightarrow K$ **do**
11:             /* Sample random noise. */
12:             $\mathsf{F}^0 \sim \mathcal{N}(0, I)$
13:             /* Sample a random timestep. */
14:             $t \sim [0, 1]$
15:             /* Calculate noisy input. */
16:             $\mathsf{F}^t_{\text{noisy}} \leftarrow \alpha_t x_{\text{img}} + \beta_t \mathsf{F}^0_i$
17:             /* Calculate first-order ground-truth. */
18:             $\mathsf{F}^t_{\text{first}} \leftarrow \alpha'_t x_{\text{img}} + \beta'_t \mathsf{F}^0_i$
19:             /* Calculate second-order ground-truth. */
20:             $\mathsf{F}^t_{\text{second}} \leftarrow \alpha''_t x_{\text{img}} + \beta''_t \mathsf{F}^0_i$
21:             /* Generate condition with Transformer. */
22:             $\widehat{\mathsf{Y}} \leftarrow \mathsf{TF}(x_{\text{img}})$
23:             /* Predict first-order with FM. */
24:             $\widehat{\mathsf{F}}^t_{\text{first}} \leftarrow \mathsf{FM}_{\text{first}}(\mathsf{F}^t_{\text{noisy}}, \widehat{\mathsf{Y}})$
25:             /* Predict second-order with FM. */
26:             $\widehat{\mathsf{F}}^t_{\text{second}} \leftarrow \mathsf{FM}_{\text{second}}(\mathsf{F}^t_{\text{noisy}}, \widehat{\mathsf{Y}})$
27:             /* Caculate loss. */
28:             $\ell_c \leftarrow \|\widehat{\mathsf{F}}^t_{\text{first}} - \mathsf{F}^t_{\text{first}}\|_2^2 + \|\widehat{\mathsf{F}}^t_{\text{second}} - \mathsf{F}^t_{\text{second}}\|_2^2$
29:             $\ell \leftarrow \ell + \ell_c$
30:             /* Downsample $x_{\text{img}}$ for next iteration. */
31:             $x_{\text{img}} \leftarrow \Phi_{\text{down}} x_{\text{img}}$
32:         **end for**
33:         /* Optimize parameter $\theta$ with $\ell$. */
34:         $\theta \leftarrow \nabla_\theta \ell$
35:     **end while**
36:     **return** $\theta$
37: **end procedure**

---

---

**Algorithm 2** High-Order FlowAR Inference

---
1: **procedure** HOFARINFERENCE($c_{\text{input}}$)
2:     /* $c_{\text{input}}$ denotes the condition embedding used for generation. */
3:     /* Init the Transformer input $x$ with $c_{\text{input}}$. */
4:     $x \leftarrow c_{\text{input}}$
5:     /* Init the $x_{\text{img}}$ with random noise. */
6:     $x_{\text{img}} \leftarrow \mathcal{N}(0, I)$
7:     /* Inference through $K$ pyramid scales. */
8:     **for** $i = 1 \rightarrow K$ **do**
9:         /* Pass through the Transformers TF. */
10:         $\widehat{\mathsf{Y}} \leftarrow \mathsf{TF}(x)$
11:         /* Extract last $i * i$ tokens from $\mathsf{Y}$ as the condition embedding. */
12:         $x_{\text{cond}} \leftarrow \mathsf{Y}[..., -i * i :]$
13:         /* Generate first-order with $\mathsf{FM}_{\text{first}}$. */
14:         $\widehat{y}_{\text{first}} \leftarrow \mathsf{FM}_{\text{first}}(x_{\text{cond}}, x_{\text{img}})$
15:         /* Generate second-order with $\mathsf{FM}_{\text{second}}$. */
16:         $\widehat{y}_{second} \leftarrow \mathsf{FM}_{\text{second}}(x_{\text{cond}}, x_{\text{img}})$
17:         /* Apply first and second-order terms. */
18:         $x_{\text{img}} \leftarrow x_{\text{img}} + \widehat{y}_{\text{first}} \cdot \Delta t + 0.5 \cdot \widehat{y}_{\text{second}} \cdot (\Delta t)^2$
19:         /* Upsample $x_{\text{img}}$. */
20:         $x_{\text{img}} \leftarrow \phi_{\text{up}}(x_{\text{img}})$
21:         /* Concatenate upsampled $x_{\text{img}}$ to the Transformer input. */
22:         $x \leftarrow \mathsf{Concat}(x, x_{\text{img}})$
23:     **end for**
24:     /* Return the final image */
25:     **return** $x_{\text{img}}$
26: **end procedure**

---

## 4 TECHNICAL OVERVIEW

In this section, we present the key lemmas used to prove the main theorem introduced in the previous section. Specifically, we first analyze the computational complexity of each component in auto-regressive Transformers and the Flow-Matching architecture. Then, we integrate these results to derive the overall runtime for both the Transformer and Flow-Matching components.

We begin by analyzing the runtime of the auto-regressive Transformer module.

**Lemma 4.1** (Running time for Auto-Regressive Transformer Forward)**.** *Let the auto-regressive Transformer is defined as in Definition 2.6 and that it contains $m$ attention layers. Let $x_{\text{img}} \in \mathbb{R}^{n \times n \times c}$ be the input image, where $n$ denotes the resolution and $c$ denotes the number of channels, and let $d$ denote the hidden dimension. Under these conditions, the running time for a single forward pass of the auto-regressive Transformer is*

$$O(mn^4 d).$$

*Proof.* We consider each attention block in the Transformers architecture.

For each attention block, it consists of the following three steps:

**Step 1: Generate matrices** $Q, K, V$**.**

We need to generate a query vector $q \in \mathbb{R}^d$, a key vector $k \in \mathbb{R}^d$ and a value vector $v \in \mathbb{R}^d$ for each pixel in the original $n \times n$ image $x_{\text{img}}$. After this step, we will have three matrices $Q, K, V \in \mathbb{R}^{n^2 \times d}$. This step takes $O(n^2 d)$ time.

**Step 2: Calculate the attention matrix.**

As defined in Definition 2.4, we need to calculate the attention matrix. It takes $O(n^4 d)$ to calculate $QK^\top \in \mathbb{R}^{n^2 \times n^2}$. It takes $O(n^4)$ time to calculate $\exp(QK^\top)$. It takes $O(n^2)$ time to calculate $D = \exp(QK^\top)\mathbf{1}_{n^2}$. It takes $O(n^2)$ to calculate the $D^{-1}$. It takes $O(n^4)$ to multiply $D^{-1}$ to each

row of $\exp(QK^\top)$. The overall running time is $O(n^4d)$. After this step, we will get the attention matrix $A \in \mathbb{R}^{n^2 \times n^2}$.

**Step 3: Calculate the final output.**

The final step is to calculate $A \cdot V$. Since $A \in \mathbb{R}^{n^2 \times n^2}$ and $V \in \mathbb{R}^{n^2 \times d}$. The running time of this step is $O(n^4d)$. Therefore, according to the above analysis, the running time for a single attention operation is $O(n^4d)$. Since there are total $m$ attention layers in the auto-regressive Transformer, the overall running time is $O(mn^4d)$. $\qquad\square$

Another crucial component of the HOFAR model is the flow-matching architecture. Following a similar approach, we analyze the computational complexity of the flow-matching model as follows:

**Lemma 4.2** (Running time for Flow-Matching Forward). *Let the auto-regressive Transformer* TF *be defined as in Definition 2.6 and that the flow-matching architecture is defined as in Definition B.4. Let $x_{\mathrm{img}} \in \mathbb{R}^{n \times n \times c}$ denote the image, where $n$ denotes the resolution and $c$ denotes the number of channels, and let $d$ denote the hidden dimension. Under these conditions, we can show that the running time for a single forward pass of the flow-matching architecture is*

$$O(n^4d^2).$$

*Proof.* Since the input of the flow-matching is the output of the auto-regressive Transformer, which is $\mathsf{TF}(x_{\mathrm{img}}) \in \mathbb{R}^{n^2 \times n^2 \times d}$. According to the definition of flow-matching architecture (Definition B.4), it consists of three operations: one MLP layer, one attention layer, and one MLP layer. For the first layer, the MLP layer, the running complexity is $O(n^2d^2)$. For the second layer, the attention layer, according to the proof of Lemma 4.1, the running time for this layer is $O(n^4d)$. For the third layer, the MLP layer, the running complexity is $O(n^2d^2)$. Therefore, the overall running time for the flow-matching is $O(n^4d^2)$. $\qquad\square$

With the runtime analysis of both the Transformer and Flow-Matching modules completed, we now proceed to analyze the training procedure of the HOFAR model. In the following proof, we break down the training process step by step and derive the overall computational complexity at the end.

**Lemma 4.3** (Running time for HOFAR training). *Suppose that the auto-regressive Transformer is defined as in Definition 2.6 and contains $m$ attention layers. Let the flow-matching architecture be defined as in Definition B.4, and assume that the HOFAR training process is described in Algorithm 1. Furthermore, suppose that HOFAR consists of $k$ pyramid frames, let $d$ denote the hidden dimension, and let $x_{\mathrm{img}} \in \mathbb{R}^{n \times n \times c}$ denote the image with resolution $n$ and $c$ channels. Then, the running time of the training procedure of HOFAR is*

$$O(kmn^4d^2).$$

*Proof.* We first consider the running time for each pyramid frame in the training loop (Line 11 to 31 in Algorithm 1). In each loop, we first consider time complexity for the preparation of essential variables (Line 15 to 20). Since the dimension of each variable in this process is $n \times n \times d$, the running complexity for the preparation process is $O(n^2d)$. Then, we consider the process of generating condition embeddings with Transformer (Line 22). According to Lemma 4.1, the running time for this process is $O(mn^4d)$. Next, according to Lemma 4.2, the prediction process of the flow-matching models takes $O(n^4d^2)$ time. Finally, the loss calculation step (Line 28) takes $O(n^2d)$ time.

Therefore, according to all the analysis mentioned above, the running time for each iteration is $O(mn^4d^2)$. Since there are total $k$ pyramid frames, the overall running time for the training process is $O(kmn^4d^2)$. $\qquad\square$

Following a similar procedure, we can have the running complexity analysis for the inference procedure as follows:

**Lemma 4.4** (Running time for HOFAR inference). *Let the auto-regressive Transformer be defined as in Definition 2.6 and contain $m$ attention layers, that the flow-matching architecture is defined as in Definition B.4, and that the HOFAR inference process is described in Algorithm 2. Also, suppose there are $k$ pyramid frames in HOFAR and let $d$ denote the hidden dimension. Under these conditions, the running time of the HOFAR inference procedure is $O(kmn^4d^2)$.*

*Proof.* We begin with considering each loop in $k$-th inference (Line 9 to Line 22). For each loop, according to Lemma 4.1, the Transformer forward pass (Line 10) takes $O(mn^4d)$ time. Next, according to Lemma 4.2, the flow-matching prediction process (Line 13 - 16) takes $O(n^4d^2)$ time. Finally, the time for applying the predicted gradient on image (Line 18) takes $O(n^2d)$ time. Therefore, the overall running time for each inference loop is $O(mn^4d^2)$. Since there are total $k$ inference loops, the overall running time is $O(kmn^4d^2)$. $\qquad\square$

Combining all the analyses discussed above, we can directly arrive at our final theorem (Theorem 3.1).

## 5 EXPERIMENTS

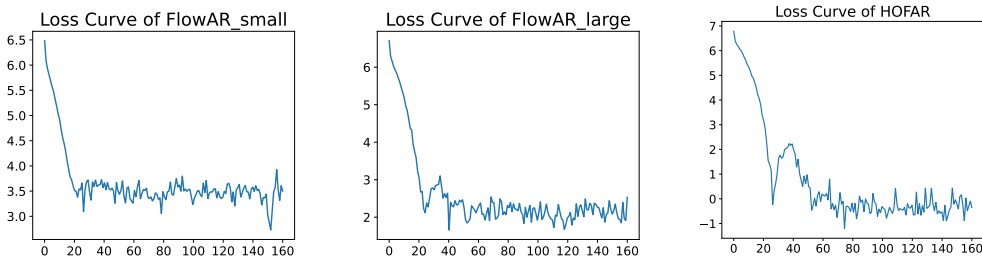

Figure 1: Loss curve of FlowAR-small (**Left**), loss curve of FlowAR-large (**Middle**) and loss curve of HOFAR (**Right**).

In Section 5.1, we introduce the setting we used in our experiments. In Section 5.2, we present the loss curve of the various models. In Section 5.3, we present visualization examples produced by the FlowAR-small, FlowAR-large and HOFAR, highlighting differences in color accuracy and generation quality on CIFAR-10 images.

### 5.1 EXPERIMENT SETUP

In FlowAR-small, we employ an embedding with three dimensions, and its Autoregressive component is configured with a 1024-dimensional feature space across a depth of 2 layers. Additionally, the flow-matching component is realized through a single hidden layer MLP operating with a step increment of 25. By comparison, FlowAR-large distinguishes itself by utilizing an eight dimension embedding and extending the Autoregressive feature dimension to 1536, while retaining the same configuration for the remaining components as in FlowAR-small. In the case of HOFAR, an embedding of dimension three is similarly adopted, paired with a 1024 dimension Autoregressive component structured over two layers, and a single-hidden-layer MLP is again employed for flow-matching with 25 steps. All three models were evaluated on the CIFAR-10 dataset, with analysis restricted to 8 classes due to computational constraints. All models above use AdamW optimizer with 0.0001 learning rate. In all experiments, the models were optimized by minimizing the sum of squared errors (SSE), and performance assessment during testing was based on the Euclidean distance metric. Regarding the target transport trajectory, we integrated the VP ODE framework as described in Liu et al. (2022), represented by $x_t = \alpha_t x_0 + \beta_t x_1$. Here, $\alpha_t$ is defined as

$$\alpha_t := \exp\left(-\frac{1}{4}a(1-t)^2 - \frac{1}{2}b(1-t)\right),$$

and $\beta_t$ is determined by $\sqrt{1 - \alpha_t^2}$, with the hyperparameters fixed at $a = 19.9$ and $b = 0.1$. During generation, the eight distinct training labels were provided as input, and a consistent $cfg$ value of 4.3 was maintained for all three models.

### 5.2 LOSS FUNCTION CURVE

Now, we present the testing loss curves of the various models during training, providing insights into their convergence behavior and learning dynamics. Figure 1 illustrates the loss for FlowAR-

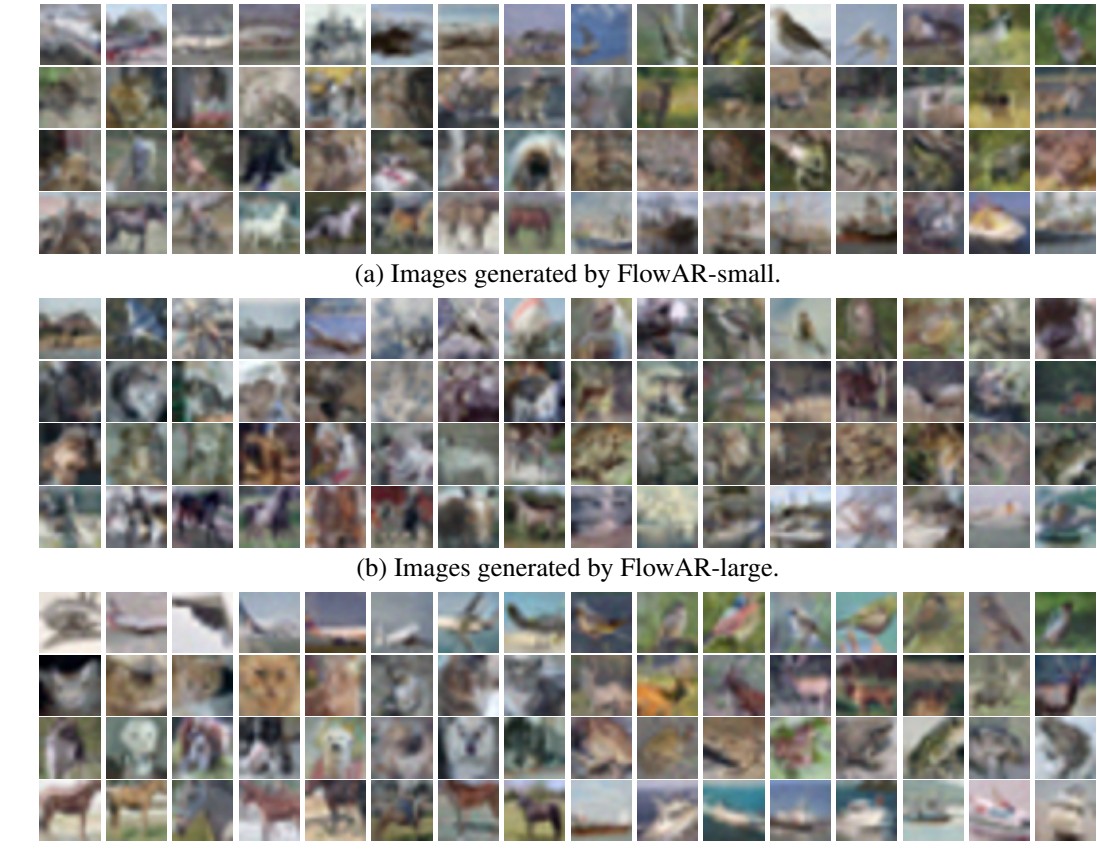

(a) Images generated by FlowAR-small.

(b) Images generated by FlowAR-large.

(c) Images generated by HOFAR.

Figure 2: Comparison of 32*32 CIFAR-10 images generation by FlowAR-small (**first four lines**), FlowAR-large (**second four lines**) and HOFAR (**last four lines**). For better looking, we put higher-resolution version of Figure 3, Figure 4 and Figure 5 here.

small, FlowAR-large, and our HOFAR, with the respective model parameter counts being 170.70M, 222.72M, and 212.44M.

## 5.3 VISUALIZATION COMPARISON

As Figure 2 shows, the visualization instances generated by the FlowAR-small, FlowAR-large and HOFAR models are delineated in this study. Each model uses the same prompt at the corresponding position.

## 6 CONCLUSION

In this work, we presented High-Order FlowAR (HOFAR), a novel framework that integrates high-order dynamics into flow-matching-based auto-regressive generation. By modeling higher-order interactions, HOFAR enhances the ability to capture complex dependencies, leading to improved realism, coherence in generative tasks. Our theoretical analysis demonstrates that HOFAR maintains computational efficiency while benefiting from high-order. Empirical evaluations further validate the superiority of HOFAR over existing auto-regressive generative models. These contributions highlight the potential of incorporating high-order dynamics into generative frameworks, paving the way for more advanced generative models in the future.

ETHIC STATEMENT

This paper does not involve human subjects, personally identifiable data, or sensitive applications. We do not foresee direct ethical risks. We follow the ICLR Code of Ethics and affirm that all aspects of this research comply with the principles of fairness, transparency, and integrity.

REPRODUCIBILITY STATEMENT

We ensure reproducibility on both theoretical and empirical fronts. For theory, we include all formal assumptions, definitions, and complete proofs in the appendix. For experiments, we describe model architectures, datasets, preprocessing steps, hyperparameters, and training details in the main text and appendix.

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

# Appendix

**Roadmap.** In Section A, we introduce related work. In Section B, we provide a formal mathematical definition of flow modeling and present the implementation of the flow-matching architecture. In Section C, we analyze the strengths and limitations of the High-Order FlowAR (HOFAR) framework. In Section D, we exhibit some result obtained from the experiments.

## A  RELATED WORK

### A.1  FLOW-BASED AND DIFFUSION-BASED GENERATIVE MODELS

Flow-based and diffusion-based generative models have demonstrated significant potential in image and video generation tasks Ho et al. (2020); Hoogeboom et al. (2023); Li et al. (2024b); Gu et al. (2024); Jain et al. (2024); Xu et al. (2024); Liu et al. (2024). Among these, Latent Diffusion Models (LDM) Rombach et al. (2022) have emerged as a particularly powerful approach, especially in the domain of text-to-image synthesis. Recent advancements, such as Stable Diffusion V3 Esser et al. (2024), have integrated flow-matching techniques as an alternative strategy to further improve generation quality and enhance the photorealism of synthesized images. Moreover, a growing body of research Jin et al. (2024); Wang et al. (2024b; 2023; 2024a) has highlighted the potential of combining the strengths of diffusion models and flow-matching models to achieve even greater generation fidelity. In this context, we acknowledge several influential works in flow-matching and diffusion-based generation Hu et al. (2022); Song et al. (2025); Dalva & Yanardag (2024); Huang et al. (2024); Wu et al. (2024); Cao et al. (2025a); Liang et al. (2025); Shen et al. (2024); Li et al. (2024a); Hu et al. (2025); Cao et al. (2025c); Ke et al. (2025b); Cao et al. (2025b); Ke et al. (2025a); Liang et al. (2024b;c); Gong et al. (2025), which have greatly inspired our research.

### A.2  HIGH-ORDER DYNAMIC SUPERVISION

High-order dynamics are often overlooked in the research community, despite their critical role in modeling target distributions—such as image or video distributions—with greater accuracy and effectiveness. Current research primarily explores high-order dynamics within gradient-based methods. For example, solvers Djeumou et al. (2022); Hong et al. (2024) and regularization frameworks Kelly et al. (2020); Finlay et al. (2020) for neural ordinary differential equations (neural ODEs) Chen et al. (2018); Grathwohl et al. (2018) frequently leverage higher-order derivatives to enhance performance Rout et al. (2024); Chen et al. (2025b;a). Beyond machine learning, the study of higher-order temporal Taylor methods (TTMs) has been extensively applied to solving both stiff Chang & Corliss (1994) and non-stiff Chang & Corliss (1994); Corliss & Chang (1982) systems, demonstrating their broad utility in computational mathematics.

**Roadmap.** This paper is organized as follows: Section 2 introduces the fundamental notations used throughout the paper and provides formal definitions for each module in the proposed model. In Section 3, we present the training and inference algorithms for our HOFAR model, along with an analysis of its computational efficiency. In Section 4, we delve into the technical details and methodologies employed to prove our formal theorem. In Section 5, we conduct an empirical evaluation of the HOFAR model, showcasing its effectiveness and robustness in image generation tasks. Finally, in Section 6, we summarize the key contributions of this paper and provide concluding remarks.

## B  FLOW MATCHING ARCHITECTURE

We begin by outlining the concept of velocity flow in the flow-matching architecture. This section introduces the foundational definitions and components necessary to understand the flow-matching model.

**Definition B.1** (Flow). *If the following conditions hold:*

- *Let $\mathsf{X} \in \mathbb{R}^{h \times w \times c}$ denote the input tensor, where $h, w, c$ represent height, width, and the number of channels, respectively.*

- *Let $K \in \mathbb{N}$ denote the scales number.*

- *For $i \in [K]$, let $\mathsf{F}_i^0 \in \mathbb{R}^{(h/r_i) \times (w/r_i) \times c}$ denote the noise tensor with every entry sampled from $\mathcal{N}(0, 1)$.*

- *For $i \in [K]$, let $\widehat{\mathsf{Y}}_i \in \mathbb{R}^{(h/r_i) \times (w/r_i) \times c}$ denote the token maps generated by autoregressive transformer as defined in Definition 2.6.*

*Then, we define the flow model supports the following two operations:*

- **Interpolation:** *For timestep $t \in [0, 1]$ and scale $i$,*

$$\mathsf{F}_i^t := t\widehat{\mathsf{Y}}_i + (1 - t)\mathsf{F}_i^0$$

*which describes a linear trajectory between the noise $\mathsf{F}_0^i$ and target tokens $\widehat{\mathsf{Y}}_i$.*

- **Velocity Field:** *The time derivative of the flow at scale $i$ is given by*

$$\mathsf{V}_i^t := \frac{\mathrm{d}\mathsf{F}_i^t}{\mathrm{d}t} = \widehat{\mathsf{Y}}_i - \mathsf{F}_i^0.$$

*This velocity field is constant across $t$ due to the linear nature of the interpolation.*

Before introducing the implementation of the flow-matching model, we first define two essential components: the Multi-Layer Perceptron (MLP) layer and the Layer Normalization (LN) layer. These components are critical for constructing the flow-matching architecture.

**Definition B.2** (MLP Layer). *If the following conditions hold:*

- *Let $\mathsf{X} \in \mathbb{R}^{h \times w \times c}$ denote the input tensor, where $h, w, c$ represent height, width, and the number of channels, respectively.*

- *Let $W \in \mathbb{R}^{c \times d}$ denote the weight matrix and $b \in \mathbb{R}^{1 \times d}$ denote the bias vector.*

*We define the MLP layer as $\mathsf{Y} := \mathsf{MLP}(\mathsf{X}, c, d)$, which outputs tensor $\mathsf{Y} \in \mathbb{R}^{h \times w \times d}$ by using the following operations:*

- *Reshape $\mathsf{X}$ into a matrix $X \in \mathbb{R}^{hw \times c}$ with spatial dimensions collapsed.*

- *For all $j \in [hw]$, we apply affine transformation on each row as follows*

$$Y_{j,*} = \underbrace{X_{j,*}}_{1 \times c} \cdot \underbrace{W}_{c \times d} + \underbrace{b}_{1 \times d}$$

- *Reshape $Y \in \mathbb{R}^{hw \times d}$ into $\mathsf{Y} \in \mathbb{R}^{h \times w \times d}$.*

Next, we define the Layer Normalization layer, which is a key component for stabilizing and normalizing the inputs to the flow-matching architecture.

**Definition B.3** (Layer Normalization Layer). *If the following conditions hold:*

- *Let $\mathsf{X} \in \mathbb{R}^{h \times w \times c}$ denote the input tensor, where $h, w, c$ represent height, width, and the number of channels, respectively.*

*We define the layer normalization as $\mathsf{Y} := \mathsf{LN}(\mathsf{X})$, which computes $\mathsf{Y}$ through the following steps*

- *Reshape $\mathsf{X}$ into a matrix $X \in \mathbb{R}^{hw \times c}$ with spatial dimensions collapsed.*

- *For each $j \in [hw]$, we apply normalization on each row of the matrix,*

$$Y_{j,*} = (X_{j,*} - \mu_j)\sigma_j^{-1}$$

*where*

$$\mu_j := \sum_{k=1}^{c} X_{j,k}/c, \quad \sigma_j = (\sum_{k=1}^{c} (X_{j,k} - \mu_j)^2/c)^{1/2}$$

- *Reshape $Y \in \mathbb{R}^{hw \times c}$ into $\mathsf{Y} \in \mathbb{R}^{h \times w \times c}$.*

With the MLP and Layer Normalization layers defined, we now introduce the flow-matching layer, which is a core component of the FlowAR model.

**Definition B.4** (Flow Matching Architecture). *If the following conditions hold:*

- *Let $\mathsf{X} \in \mathbb{R}^{h \times w \times c}$ denote the input tensor, where $h, w, c$ represent height, width, and the number of channels, respectively.*

- *Let $K \in \mathbb{N}$ denote the number of total scales in FlowAR.*

- *For $i \in [K]$, let $\widehat{\mathsf{Y}}_i \in \mathbb{R}^{(h/r_i) \times (w/r_i) \times c}$ denote the token maps generated by autoregressive transformer defined in Definition 2.6.*

- *For $i \in [K]$, let $\mathsf{F}_i^t \in \mathbb{R}^{(h/r_i) \times (w/r_i) \times c}$ denote interpolated input defined in Definition B.1.*

- *For $i \in [K]$, let $t_i \in [0, 1]$ denote timestep.*

- *For $i \in [K]$, let $\mathsf{Attn}_i(\cdot) : \mathbb{R}^{h/r_i \times w/r_i \times c} \to \mathbb{R}^{h/r_i \times w/r_i \times c}$ denote the attention layer as defined in Definition 2.4.*

- *For $i \in [K]$, let $\mathsf{MLP}_i(\cdot, c, d) : \mathbb{R}^{h/r_i \times w/r_i \times c} \to \mathbb{R}^{h/r_i \times w/r_i \times c}$ denote the MLP layer as defined in Definition B.2.*

- *For $i \in [K]$, let $\mathsf{LN}_i(\cdot) : \mathbb{R}^{h/r_i \times w/r_i \times c} \to \mathbb{R}^{h/r_i \times w/r_i \times c}$ denote the layer norm layer as defined in Definition B.3.*

*Then we define the flow-matching architecture as $\mathsf{F}_i''^{t_i} := \mathsf{FM}_i(\widehat{\mathsf{Y}}_i, \mathsf{F}_i^{t_i}, t_i)$, which contains the following computation steps:*

- *Generate parameter conditioned on the timestep,*

$$\alpha_1, \alpha_2, \beta_1, \beta_2, \gamma_1, \gamma_2 := \mathsf{MLP}_i(\widehat{\mathsf{Y}}_i + t_i, c, 6c)$$

- *Apply attention mechanism,*

$$\mathsf{F}_i'^{t_i} := \mathsf{Attn}_i(\gamma_1 \circ \mathsf{LN}(\mathsf{F}_i^{t_i}) + \beta_1) \circ \alpha_1$$

*with $\circ$ denoting Hadamard (element-wise) product.*

- *Apply MLP and LN modules,*

$$\mathsf{F}_i''^{t_i} := \mathsf{MLP}_i(\gamma_2 \circ \mathsf{LN}(\mathsf{F}_i'^{t_i}) + \beta_2, c, c) \circ \alpha_2$$

## C  DISCUSSION

The HOFAR framework introduces a novel approach to integrating high-order dynamics into flow-matching-based auto-regressive generation, significantly improving the modeling of complex dependencies and generation quality. However, certain limitations and future directions deserve attention. One limitation is the potential computational overhead when scaling HOFAR to extremely high-dimensional data, such as ultra-high-resolution images or long-duration videos. While HOFAR maintains theoretical efficiency, practical implementation may require further optimization to handle such scenarios. Future work could explore extending HOFAR to multi-modal generation tasks, such as joint text-video or text-3D generation, where capturing long-term coherence across modalities is critical. Furthermore, improving the interpretability of high-order dynamics through visualization or disentanglement techniques would broaden HOFAR's applicability.

## D  EMPIRICAL RESULT

In Section D.1, we compare visualizations generated by FlowAR and our HOFAR, this highlighting differences in color accuracy and relative position on CIFAR-10 images.

### D.1 VISUALIZATION EXAMPLES

We present visualization examples produced by the FlowAR-small, FlowAR-large and proposed HOFAR. Specifically, Figure 3 showcases visualizations generated by the FlowAR-small model, Figure 4 showcases visualizations generated by the FlowAR-large model, whereas Figure 5 highlights visualizations created by the HOFAR model.

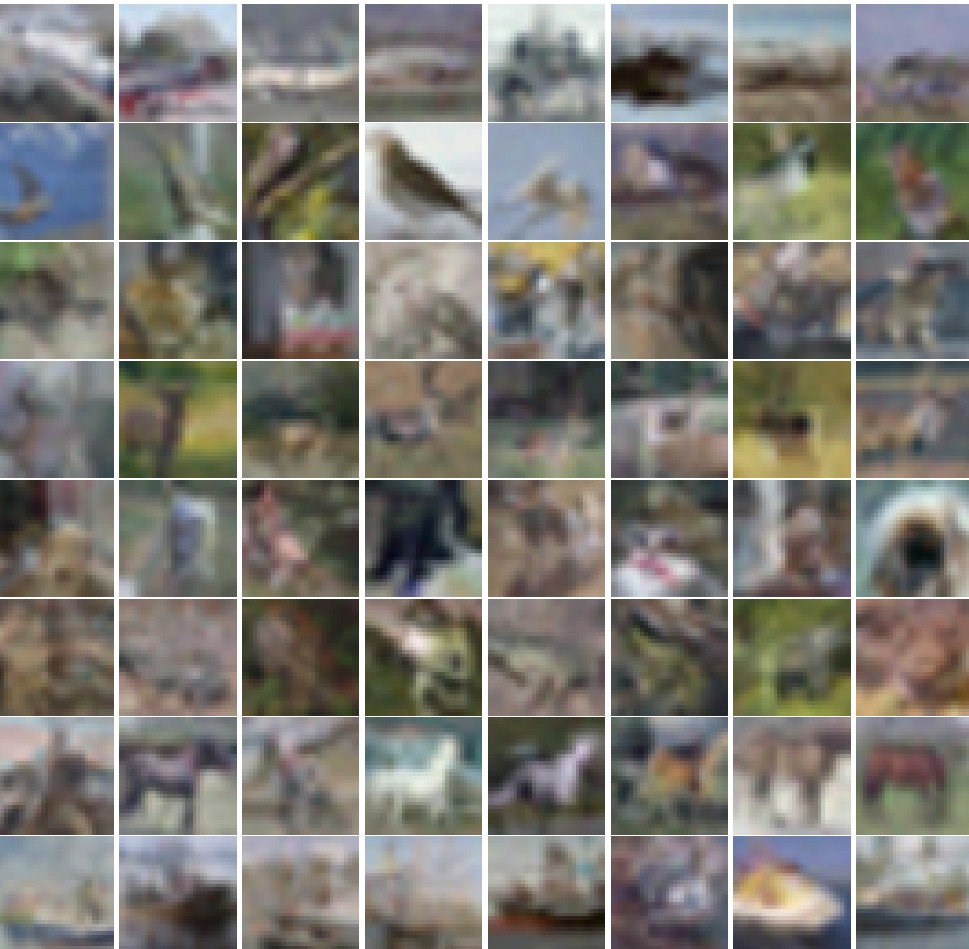

Figure 3: 64 32*32 images generated by FlowAR-small.

## LLM USAGE DISCLOSURE

LLMs were used only to polish language, such as grammar and wording. These models did not contribute to idea creation or writing, and the authors take full responsibility for this paper's content.

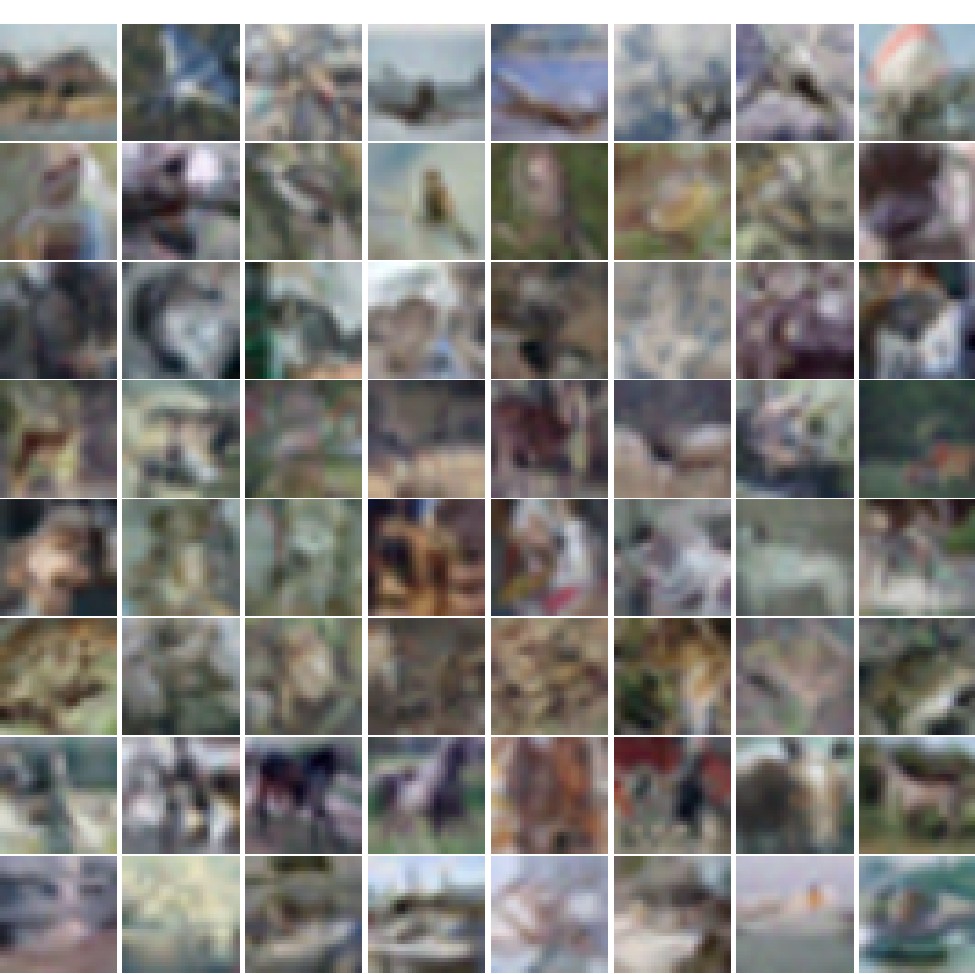

Figure 4: 64 32*32 images generated by FlowAR-large.

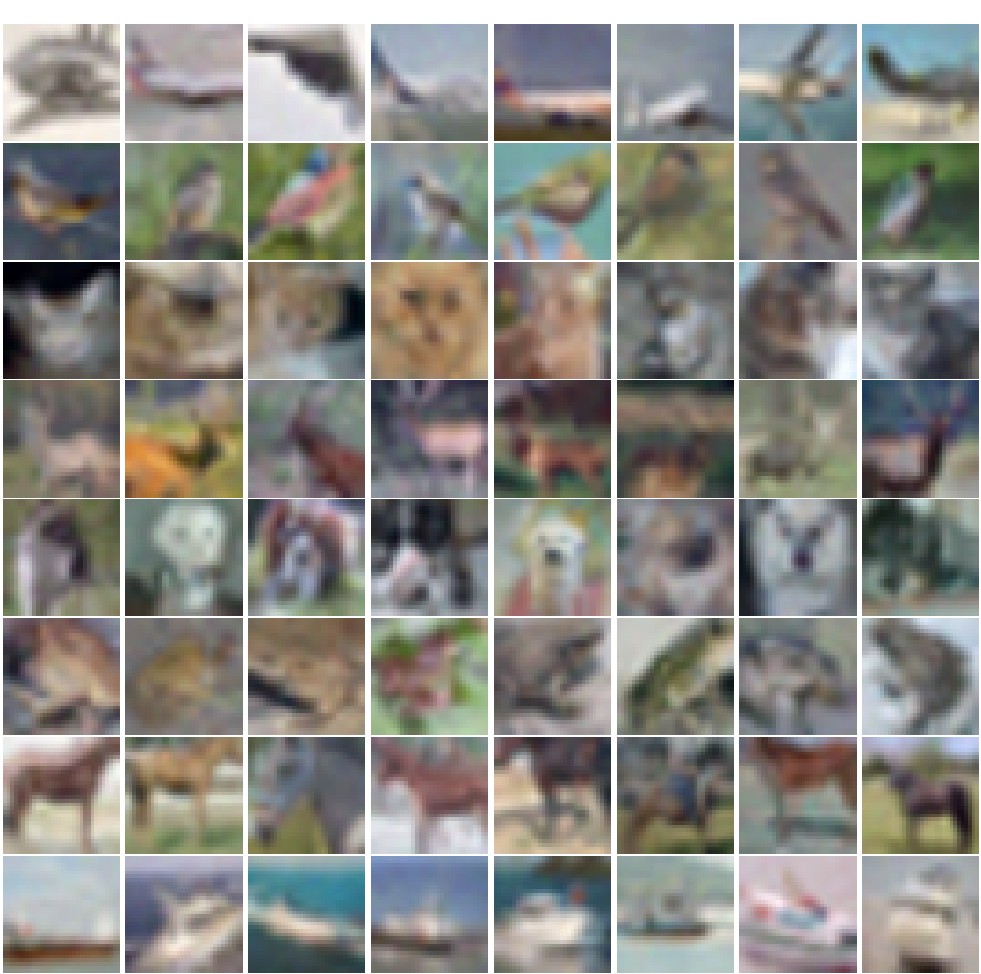

Figure 5: 64 32*32 images generated by HOFAR.

