# OpenReview forum: "HOFAR: High-Order Augmentation of Flow Autoregressive Transformers"
_ICLR.cc/2026/Conference — ICLR 2026 Conference Withdrawn Submission_

### Official Review · Reviewer_Btf4 · 2025-10-15

**Soundness:** 1
**Presentation:** 2
**Contribution:** 1
**Rating:** 2
**Confidence:** 3

**Summary:**

This paper proposes a generative model that injects high-order dynamics into the autoregressive flow matching (FlowAR) framework. In addition to predicting first-order velocity as in flow matching, it also predicts the high-order dynamics of the flow trajectory. The authors further provide an analysis of the computational complexity of the FlowAR framework. The proposed model is evaluated on CIFAR-10 and shows blurry generation results.

**Strengths:**

1. The idea of modeling high-order flow dynamics is reasonable, as the flow trajectories are usually curved due to the intersection of flow trajectories. This has been a major problem for the efficiency of flow generative models, and solving it could have a significance impact.

**Weaknesses:**

1. Poor experimental results. The qualitative results on CIFAR-10 look extremely blurry, and not comparable to the standard flow matching or autoregressive models. This casts doubt on the validity of the proposed generative model. The authors should further analyze the method design or debug the code to achieve reasonable generation performance.
2. Lack of quantitative comparisons. The authors only show the loss curve to demonstrate its convergence, without any quantitative metrics regarding its generation quality. The authors could consider metrics such as FID and IS and provide a more comprehensive comparison on CIFAR-10, CIFAR-100, and ImageNet datasets.
3. Lack of comparison to high-order solver. Instead of performing high-order flow matching, one could directly apply high-order ODE solvers to the curved flow trajectory. This has been extensively validated by DPM-Solver [Lu'22] and more recently by the Flow-DPM-Solver in SANA [Xie'24]. The authors should compare with these alternative methods to validate the advantages of the proposed method.
4. Computational efficiency is not justified. The computational complexity analysis is highly specific to the adopted FlowAR framework and does not provide sufficient insight regarding its advantages over alternative generative modeling paradigms. Furthermore, the high-order modeling design is expected to increase computational overhead and reduce efficiency in terms of computational complexity.

**Questions:**

1. Does HOFAR offer any empirical advantages over conventional diffusion models or flow-based models? For instance, it would be helpful the authors compared their method with EDM [Karras'22] and FM [Lipman'22] to clarify this point.
2. How does modeling high-order velocity in flow matching differ from directly applying high-order ODE solvers during inference? A potential experiment is to compare HOFAR with Flow-DPM-Solver [Lu'22, Xie'24].

---

> ### Author Response · Authors · 2025-12-01
>
> Thank you for your thoughtful feedback. Your comments are very helpful and much appreciated. We will address these in the next version.

---

### Official Review · Reviewer_pDME · 2025-10-31

**Soundness:** 2
**Presentation:** 2
**Contribution:** 2
**Rating:** 4
**Confidence:** 4

**Summary:**

This paper addresses the limitation of existing flow-matching autoregressive models which only model first-order dynamics of the generative process. The author(s) propose High-Order FlowAR called as HOFAR. It is a framework that augments a FlowAR-style image generator with high-order supervision. Specifically, by training the model to predict not only the first-order field of the data trajectory but also the second-order field during generation. A Transformer-based autoregressive module produces conditioning embeddings and two flow-matching networks then predict the first-order and second-order flows at each time step. During inference, the sampler uses a second-order update to more accurately step through the generative process. The paper’s contributions include: (1) introducing HOFAR as a novel integration of high-order dynamics into flow-matching autoregressive generation, (2) a theoretical analysis showing that incorporating second-order terms does not significantly increase computational complexity, and (3) empirical results demonstrating improved image generation quality and coherence compared to first-order FlowAR baselines.

**Strengths:**

1. HOFAR explicitly incorporates second-order trajectory information into this framework. Prior works like Ren et al., 2024 and Hui et al., 2025 combined flow matching with Transformers but were limited to first-order dynamics.

2. Adding the second-order flow network incurs only a marginal overhead in computation, which is supported by the fact that HOFAR’s 212.4M parameter count is comparable to the first-order 222.7M large model. The methodology seems sound. The authors train HOFAR by matching both first and second derivative fields with ground truth values derived from the known continuous-time trajectory and use both predictions to advance the sampler.

3. The inclusion of pseudocode algorithms, i.e., Algorithm 1 and Algorithm 2 for HOFAR’s training and inference procedure is helpful for understanding the implementation details. Important components like the multi-scale VAE preprocessing and the transformer architecture are described with definitions and notation, which makes the technical content easily understandable. Overall, the exposition is logical.

**Weaknesses:**

1. There is no theoretical analysis of why high-order supervision must improve generation or any formal guarantee on model expressiveness or stability. For e.g., one might expect a theorem about approximation error or convergence improvements with second-order terms but the paper does not provide this.

2. Essentially, HOFAR takes the FlowAR/ARFlow architecture and adds an extra MLP to predict the second derivative and a corresponding term in the loss function. The overall framework, i.e., a multi-scale VAE encoder, an autoregressive Transformer generating conditioning tokens, and a flow-matching decoder remains the same as in FlowAR. The main difference is supervising an additional term during training and using a second-order integrator during sampling. This idea, while sensible, might be seen as an incremental improvement. The paper does not discuss in depth whether alternative approaches to high-order modeling were possible or considered.

3. The critical one to me, the absence of results on even a toy video dataset or a sequence modeling task is missing. This could help validate HOFAR’s core premise about long-term dependencies. In addition, the evaluation focuses on qualitative comparisons and loss curves; common quantitative metrics for generation (like FID or Inception Score) are not reported, which makes it harder to objectively gauge the improvement in distribution quality.

4. The submission could be stronger with more extensive ablation studies or diagnostics.. Without ablation experiments, for e.g,  varying the number of integration steps, or turning off the second-order term, it’s hard to pinpoint why HOFAR performs better. Is it purely the additional information, or does it act as a regularizer, etc.? and how robust the approach is.

**Questions:**

Have you considered or attempted applying HOFAR to domains with pronounced temporal or spatial structure, such as video generation or 3D shape generation? The motivation emphasizes long-term coherence for e.g. in video, as a key benefit of high-order modeling, but the paper only shows static image results. What challenges do you anticipate in extending HOFAR to videos or other sequential data, and do you have plans or preliminary results in those areas?


Is the improvement from second-order supervision consistent across different datasets or settings? For e.g., if one were to apply HOFAR to a higher-resolution image dataset or to a different generation task, would you expect similar gains? Moreover, how sensitive is HOFAR’s performance to hyperparameters like the integration step size, noise schedule, or the weighting of the second-order loss term?

---

> ### Author Response · Authors · 2025-12-01
>
> Thank you for your thoughtful feedback. Your comments are very helpful and much appreciated. We will address these in the next version.

---

### Official Review · Reviewer_hrSm · 2025-10-31

**Soundness:** 3
**Presentation:** 3
**Contribution:** 3
**Rating:** 4
**Confidence:** 3

**Summary:**

The paper proposes \emph{HOFAR}, an extension to FlowAR that augments an autoregressive flow-matching generator with explicit high-order supervision along the generation trajectory. Concretely, the model trains two flow modules to predict first- and second-order derivatives of the transport path and then fuses them at inference time through a Taylor-style update to advance the state. The submission also sketches a complexity analysis arguing that training/inference scales with the number of pyramid scales, attention layers, image side length, and hidden size. Empirical evidence is limited to CIFAR-10 at $32{\times}32$ using FlowAR-small/large as baselines, with qualitative samples and loss curves.

While the idea of supervising higher-order dynamics in autoregressive flow matching is intuitive and potentially useful, the work feels preliminary for CVPR. The experimental scope is narrow (CIFAR-10 only), and there are no standard quantitative metrics (e.g., FID, IS, precision/recall, CLIP-based scores) or statistical tests to substantiate claims of improved coherence or generalization. Key ablations are missing (second-order vs.\ first-order only, effect of step size $\Delta t$, number of pyramid scales), and there is no wall-clock or memory profiling to validate the “marginal overhead” claim. Theoretical contributions are light: the complexity discussion mostly restates quadratic attention costs without analyzing approximation error or stability benefits of the second-order term. Important implementation details (e.g., attention configuration, normalization, masking) are under-specified, and the training objective reduces to an SSE on derivative targets without justification of statistical optimality.

**Strengths:**

The paper cleanly augments FlowAR with explicit high-order supervision (first/second derivatives) and presents it as a general recipe for AR flow models, with a succinct statement of novelty in the abstract.

**Weaknesses:**

1. All results are on CIFAR-10 at 32×32, and even then the analysis is restricted to 8 classes due to computational constraints—so generality is unclear.

2. The paper reports loss curves and an Euclidean-distance test metric, but no FID/IS/precision–recall/CLIP-FID, making it impossible to gauge perceptual quality or distribution fidelity.

3. The method minimizes SSE on derivative targets; there’s no discussion of statistical optimality or robustness.

4. There’s no isolation of contributions (e.g., first- vs. second-order supervision, step size $\delta$ t, number of pyramid scales), despite these being central design choices; Section 5 lists only curves and samples.

**Questions:**

You only show loss curves and visuals; why are there no standard generative metrics (FID/IS/precision–recall)? Please add them and explain how they align (or not) with your observed loss trends.

---

> ### Author Response · Authors · 2025-12-01
>
> Thank you for your thoughtful feedback. Your comments are very helpful and much appreciated. We will address these in the next version.

---

### Official Review · Reviewer_aEhC · 2025-11-01

**Soundness:** 2
**Presentation:** 2
**Contribution:** 3
**Rating:** 4
**Confidence:** 5

**Summary:**

This paper presents a new framework, High-Order FlowAR (HOFAR), designed to enhance generative models that combine flow matching with autoregressive transformers. The authors identify a key limitation in existing FlowAR models: their reliance on first-order trajectory modeling. To address this, HOFAR integrates high-order supervision, specifically second-order dynamics, into the generation process. The objective is to enable the model to capture more complex dependencies and improve the realism and coherence of the generated images.

The core of the HOFAR method involves modifying both the training and inference stages. During training, the model is supervised using two targets: the first-order velocity ($F_{first}^{t}$) and the second-order derivative ($F_{second}^{t}$). Two distinct flow matching networks ($FM_{first}$ and $FM_{second}$) are employed to predict these two dynamic terms. The overall loss function is a sum of the squared errors for both the first and second-order predictions. For inference, the model utilizes both predictions in a second-order Taylor-like update step: $x_{img} \leftarrow$ $x_{img}$ + \hat{y}_{first} \cdot \Delta t + 0.5 \cdot \hat{y}_{second} \cdot (\Delta t)^{2}$.

The paper provides a theoretical analysis of computational complexity, showing that both training and inference for HOFAR operate at $O(kmn^{4}d^{2})$. The authors assert that this integration of high-order information introduces only a marginal increase in computational cost compared to the baseline. Experimental evaluations on the CIFAR-10 dataset (using 8 classes) demonstrate that HOFAR (212.44M parameters) produces higher-quality images than the FlowAR-large model (222.72M parameters), suggesting an improved trade-off between parameter count and generation fidelity.

**Strengths:**

+ HOFAR systematically addresses the limitation of first-order trajectory modeling in $FlowAR$ models by integrating high-order dynamics supervision.

+ The paper includes a clear theoretical analysis demonstrating that the integration of high-order dynamics maintains computational efficiency for both training and inference, a crucial aspect for scaling generative models (theorem 3.1).

+ The framework enhances the model's ability to capture complex dependencies, which is stated to improve realism, coherence, and generalization in generative tasks.

+ The paper is clear in defining its complex components using formal mathematical definitions for the transformer and flow-matching architectures (section 2 and appendix B).

**Weaknesses:**

+ The empirical evaluation is currently incomplete. The paper lacks standard quantitative metrics essential for assessing image generation quality, such as FID (Fréchet Inception Distance), Inception Score, or SSIM. Solely relying on visual comparison (figure 2) and loss curves (figure 1) is insufficient to fully validate the superiority of HOFAR over FlowAR-large.  Especially since FlowAR-large has a higher parameter count (222.72M vs. 212.44M for HOFAR).

+ The experiments are only performed on the CIFAR-10 dataset (restricted to $8$ classes). While useful for proof-of-concept, to demonstrate true generalization and the benefit of high-order dynamics—especially for "long-term coherence" and "complex dependencies"—evaluation on higher-resolution images (e.g., ImageNet, CelebA-HQ) or, ideally, a video generation task (given the mention of long-term coherence) is crucial.

+ The full flow-matching architecture definition (Definition B.4) is relegated to the appendix, while its time complexity is critical to the main theorem's proof. This should be made more accessible or summarized in the main paper.

+ There are no ablation studies on the order of supervision (e.g., comparing first-order only vs. first- and second-order, or varying the magnitude of the second-order term). This is vital to truly isolate the contribution of the high-order dynamics.

**Questions:**

- Please provide standard quantitative generation metrics (FID, Inception Score) comparing HOFAR, FlowAR-small, and FlowAR-large on the CIFAR-10 dataset. Without these, the empirical validation of "improved generation quality" is weak.

- Can the authors provide an ablation study quantifying the individual contribution of the second-order term? For example, compare FlowAR (first-order only) against HOFAR (first-order plus $\lambda \cdot$ second-order), where $\lambda$ is a weight. This would validate the core claim that high-order dynamics provide a measurable benefit.

- The introduction strongly suggests high-order dynamics are important for long-term coherence in tasks like video generation. Given this, can the authors elaborate on future plans or preliminary results on a video dataset to substantiate the model's relevance to this domain?

- In $Lemma~4.1$, the attention matrix calculation is stated to take $O(n^{4}d)$ and $O(n^{4})$ for specific steps. Since $n$ is the image resolution (e.g., $32\times 32$), the quadratic dependence on $n^{2}$ (the number of tokens) is expected. Please clarify if this complexity includes the cost of the multi-scale autoregressive process, as the number of tokens $\sum_{j=1}^{i}h/r_{j}\cdot w/r_{j}$ grows, which could potentially dominate. Is the presented $O(mn^{4}d)$ the worst-case complexity across all scales?

---

> ### Author Response · Authors · 2025-12-01
>
> Thank you for your thoughtful feedback. Your comments are very helpful and much appreciated. We will address these in the next version.

---

### Note · Authors · 2025-12-01

I have read and agree with the venue's withdrawal policy on behalf of myself and my co-authors.